# Research on Tractor Optimal Obstacle Avoidance Path Planning for Improving Navigation Accuracy and Avoiding Land Waste

Hongtao Chen [1,2,3,*] , Hui Xie [1], Liming Sun [2,3] and Tansu Shang [2,3]

1 School of Mechanical Engineering, Tianjin University, Tianjin 300350, China; xiehui@tju.edu.cn
2 State Key Laboratory of Intelligent Agricultural Power Equipment, Luoyang 471039, China; sunliming@ytogroup.com (L.S.); stswmhluoyang@163.com (T.S.)
3 Luoyang Tractor Research Institute Co., Ltd., Luoyang 471039, China
* Correspondence: chtao@tju.edu.cn

**Abstract:** Obstacle avoidance operations of tractors can cause parts of land to be unavailable for planting crops, which represents a reduction in land utilization. However, land utilization is significant to the increase in agricultural productivity. Traditional obstacle avoidance path planning methods mostly focus on automatic tractor navigation with small errors, ignoring the decrease in land utilization due to obstacle avoidance operations. To address the problem, this paper proposed an obstacle avoidance path planning method based on the Genetic Algorithm (GA) and Bezier curve. In this paper, a third-order Bezier curve was used to plot the obstacle avoidance path, and the range of control points for the third-order Bezier curve was determined according to the global path and the location of the obstacle. To target the navigation error and land utilization problems, GA was used to search for the optimal point from the selection range of the control point under multiple constraints for automatic tractor navigation such as the obstacle collision avoidance, the minimum turning radius, and the maximum turning angle. Finally, the optimal obstacle avoidance path was determined based on the selected control points to minimize the navigation error and maximize land utilization. The algorithm proposed in this paper was compared with existing methods and the results showed that it has generally favorable performance on obstacle avoidance path planning.

**Keywords:** tractor; obstacle avoidance path planning; genetic algorithm; Bezier curve

## 1. Introduction

Tractor automatic navigation technology is widely used in many agricultural production processes such as planting and plant protection. It is necessary to plan an obstacle avoidance path for the tractor to achieve obstacle avoidance and then return to the global path again when there are obstacles in the field. So, Obstacle Avoidance Path Planning is an important research area for Tractor Automatic Guidance Systems, and a key technology that must be solved for intelligent agriculture.

There are many theories and methods that have been proposed to plan the optimal Obstacle Avoidance Path. For example, geometric curves such as the Bezier curve [1,2] and B-spline curve [3,4] have been frequently applied to plot the obstacle avoidance path. Xi et al. proposed a third-order Bezier curve optimization method to form a continuous smooth obstacle avoidance path, and the results showed favorable tracking performance of the curve path [5]. An improved cubic B-spline optimization method for obstacle avoidance in the motion of the manipulator was proposed by Wan et al. [6]. The main idea of existing methods above is to smoothly connect intermediate path points using the cubic B-spline method.

Moreover, algorithms such as the Ant Colony Algorithm [7,8], Particle Group Algorithm [9,10], A* Algorithm [11,12], and Genetic Algorithm [13,14] are widely applied in obstacle avoidance path planning and have also been applied in different industry fields.

In the ocean current environment, for instance, Ajeil et al. proposed an obstacle avoidance path planning method for an unmanned submarine based on an improved firework-ant colony algorithm [15]. At the same time, an obstacle avoidance path planning scheme for UAVs was proposed to improve the safety level of UAVs. In this paper, the path was optimized with consideration of objective functions including the minimum trajectory length, elapsed time energy consumption, and obstacle collision avoidance constrains [16]. In order to solve the problem of obstacle avoidance path planning for mobile robots, F et al. proposed a modified genetic algorithm to plan the obstacle avoidance path based on the Bezier curve [17]. With the distance between the starting point and the end point as the optimization objective, the obstacle avoidance path was planned based on the Bezier curve by determining the control points. In the field of agricultural machinery, Inoue et al. proposed a method using machine vision to detect obstacles and then plan a feasible obstacle avoidance path [18]. In summary, existing algorithms take the navigation error as the path planning objective and the land utilization is not considered.

To address this problem, this paper proposes an avoidance path planning method based on the Genetic Algorithm (GA) and Bezier curve. In this paper, the Bezier curve is used to plot the obstacle avoidance path, and the control points that determine the geometry of the third-order Bezier curve have a selection range based on the global path and locations of obstacles. The algorithm proposed in this paper consists of 3 steps as follows: (1) The kinematics model of the tractor was established to simulate automatic navigation of the tractor. (2) The selection range of control points was obtained through data of the global path and obstacle. (3) With the goal of a small error in tractor automatic navigation and high land utilization, the optimal control point was searched from the control point selection range using GA to obtain an obstacle avoidance path that satisfies multiple constraints such as anti-collision, minimum turning radius, and kinematic constraints.

## 2. Materials and Methods

### 2.1. Obstacle Avoidance Path Model

#### 2.1.1. Bezier Curve

The third-order Bezier curve is widely used in many fields, such as the path planning of mobile robots and structural modeling. The schematic diagram of the third-order Bezier curve is shown in Figure 1 and is defined as follows:

$$[x \ y] = \begin{bmatrix} t^3 & t^2 & t & 1 \end{bmatrix} GP, t \in [0,1] \tag{1}$$

where t indicates the normalized time variable; $(p_{0,x}, p_{0,y})$, $(p_{1,x}, p_{1,y})$, $(p_{2,x}, p_{2,y})$, and $(p_{3,x}, p_{3,y})$ are coordinates of control points $P_0$, $P_1$, $P_2$, and $P_3$, respectively.

$$G = \begin{bmatrix} -1 & 3 & -3 & 1 \\ 3 & -6 & 3 & 0 \\ -3 & 3 & 0 & 0 \\ 1 & 0 & 0 & 0 \end{bmatrix},$$

$$P = \begin{bmatrix} p_{0,x} & p_{0,y} \\ p_{1,x} & p_{1,y} \\ p_{2,x} & p_{2,y} \\ p_{3,x} & p_{3,y} \end{bmatrix}$$

The first derivative and second derivative of the third-order Bezier curve are expressed as in Equation (2).

$$\begin{cases} [\dot{x} \ \dot{y}] = [3t^2 \ 2t \ 1 \ 0] GP \\ [\ddot{x} \ \ddot{y}] = [6t \ 2 \ 0 \ 0] GP \end{cases} \tag{2}$$

where $\dot{x}, \dot{y}, \ddot{x}$, and $\ddot{y}$ are the components of first and second derivatives of the point (x(t),y(t)) for the X and Y coordinates, respectively.

The curvature of the third-order Bezier curve is expressed as follows:

$$k(t) = \frac{\dot{x}\ddot{y} - \ddot{x}\dot{y}}{\left(\dot{x}^2 + \dot{y}^2\right)^{\frac{3}{2}}}$$

(3)

where k(t) represents the radius of curvature.

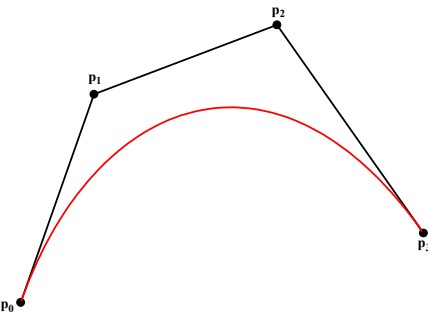

**Figure 1.** Third-order Bezier curve.

2.1.2. Obstacle Avoidance Path Model

The obstacle avoidance path was planned in the world standard latitude and longitude coordinate system (WGS84 coordinate system), and the navigation path L is a sequence of path discrete points {k0, k1, k2, . . . , ki . . . }, where ki = (loni,lati). As shown in Figure 2, the obstacle avoidance path contains two third-order Bezier curves. The shapes of third-order Bezier curves are determined based on the coordinates of control points, and the first curve is connected to the second curve through $P_{1,3} = P_{2,0}$. Therefore, the problem of obstacle avoidance path planning is transformed into the acquisition of coordinates of optimized control points [19].

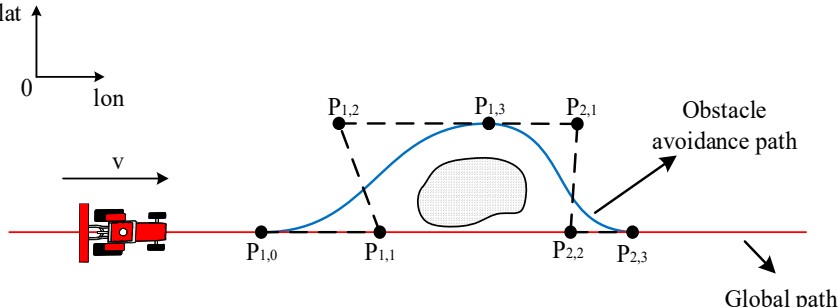

**Figure 2.** Obstacle avoidance path model.

The control point coordinates of the obstacle avoidance path have the limitation of selection range. Therefore, the optimal obstacle avoidance path is obtained by searching the range of the limit control point coordinates according to the optimal control point.

First, a kinematic model is established to simulate autonomous navigation of the tractor. Secondly, with the goal of minimizing the error of tractor automatic navigation and maximizing land utilization rate, the obstacle avoidance path is generated by searching for optimal control points, which meets the multi-constraint conditions such as the minimum turning radius, the collision avoidance, and the maximum wheel angle. Then, with the goal of small automatic tractor navigation error and less wasted land in the obstacle avoidance path, the optimal control point is searched to obtain the obstacle avoidance path that satisfies multiple constraints such as anti-collision, minimum turning radius, and kinematic constraints.

### 2.2. Tractor Kinematic Model

As shown in Figure 3, the tractor is simplified to a bicycle model [20,21]. The bicycle model is subjected to the following assumptions:

1. The tractor is a rigid body.
2. The tractor is front-wheel-steered and the left and right wheels are steered at the same angle.
3. The roll and pitch movements are ignored.
4. The lateral sliding is ignored.

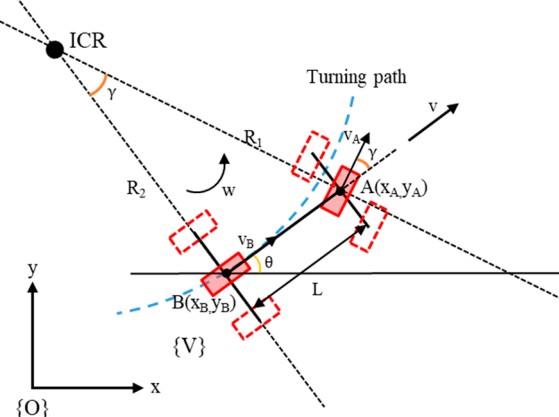

**Figure 3.** Tractor model. Note: The shape of the four-wheeled tractor is shown as a red dashed area and the approximate two-wheeled tractor is shown as a red straight area. $\gamma$ indicates the steering angle of the steering wheel (front wheel deflection) and the velocity of the rear wheel in the x-axis direction is v. The front and rear wheel axes are shown as black dashed lines, the intersection of which is the instantaneous center of rotation (ICR); the blue dashed lines indicate the steering trajectory; the distances from the front and rear wheels to the instantaneous center are $R_1$ and $R_2$, respectively. L is the length of the tractor or wheel base. A ($x_A$, $y_A$) are the rear axle coordinates, B($x_B$, $y_B$) are the front axle coordinates, $\theta$ is the traverse angle (heading angle) of the tractor, w is the traverse angular velocity of the tractor, $v_A$ is the front axle center velocity, and $v_B$ is the rear axle center velocity of the tractor.

The two-wheeled bicycle model has the rear wheels fixed to the body of the tractor and the front wheels can be turned around a horizontal axis to steer the tractor. In this paper, using the center of the rear axis as the reference point, we have the following equation:

$$\begin{cases} v = v_B \\ R = R_2 \end{cases}$$

v is the velocity of the tractor's rear wheels in the *x*-axis direction and R is the tractor's steering radius.

The tractor's poses are represented by the coordinate system {V} shown in Figure 3, with the *x*-axis being the forward direction of the tractor and the origin of the coordinates at the center of the rear wheels. The tractor 's locus is given by the generalized coordinates q = (x,y,$\theta$)$\in$S, S$\subset$SE(2) (S is the conformal space of the tractor). The velocity of the cart is defined on the basis of its velocity v in the x-direction and 0 in the y-direction, as the wheels cannot move sideways. In the cart coordinate system, this is expressed as

$$v_{\dot{x}} = v, v_{\dot{y}} = 0$$

The dotted line in the diagram indicates that the wheel cannot move in this direction, and at the intersection of the dotted lines is the instantaneous center of rotation, a reference

point on the tractor will move along an arc trajectory with the angular velocity shown in Equation (4).

$$\dot{\theta} = \frac{v}{R_1} \tag{4}$$

At the center of the rear wheel, the speed is expressed as Equation (5):

$$v = \dot{x}_B * \cos(\theta) + \dot{y}_B * \sin(\theta) \tag{5}$$

Kinematic constraints of front and rear axles (no lateral sideslip):

$$\begin{cases} \dot{x}_A * \sin(\gamma + \theta) - \dot{y}_A * \cos(\gamma + \theta) = 0 \\ \dot{x}_B * \sin(\theta) - \dot{y}_B * \cos(\theta) = 0 \end{cases} \tag{6}$$

From Equations (5) and (6), we have:

$$\begin{cases} \dot{x}_B = v\cos\theta \\ \dot{y}_B = v\sin\theta \end{cases} \tag{7}$$

Equation (8) can be obtained from the front and rear wheel geometry:

$$\begin{cases} x_A = x_B + L\cos\theta \\ y_A = y_B + L\sin\theta \end{cases} \tag{8}$$

Substituting Equations (7) and (8) into Equation (6), the angular velocity of the transverse pendulum can be obtained as:

$$\omega = \frac{v}{L}\tan\gamma \tag{9}$$

The steering radius R and the front wheel deflection angle $\gamma$ can be obtained from w and v tractor speed:

$$\begin{cases} R = v/\omega \\ \gamma = \arctan(L/R) \end{cases} \tag{10}$$

In addition, the following equations can be derived according to Figure 3:

$$\begin{cases} \dot{x} = v\cos\theta \\ \dot{y} = v\sin\theta \\ \dot{\theta} = \frac{v}{L}\tan\gamma \end{cases} \tag{11}$$

where $(x_B, y_B)$ represents the coordinate of point B; $\dot{x}_B$ and $\dot{y}_B$ are the components of the first derivatives of point B for the X and Y coordinates, respectively.

Figure 4 illustrates the tractor path tracking model. With the goal of minimizing the heading angle of the tractor, preview point C can be determined by searching for the points in the planned path, and the preview point satisfies the requirement that the distance to point $C_s$ is less than $d_1$. The tractor reaches preview point C by the desired path.

Based on Figure 4, the following results can be calculated:

$$\frac{L_d}{\sin 2\alpha} = \frac{R}{\cos \alpha} \tag{12}$$

The kinematic model of automatic tractor navigation can be obtained by combining Equations (12) and (13):

$$\theta = \arctan\left(\frac{2L\sin \alpha}{L_d}\right) \tag{13}$$

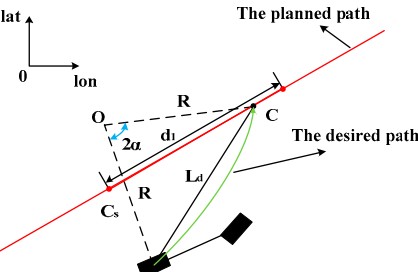

**Figure 4.** Tractor path tracking model. Note: C is a preview point on the planned path. $L_d$ is the distance between the tractor rear axle and preview point C. $2\alpha$ is the angular deviation of the tractor with respect to the preview point C. $C_S$ is the point on the planned path with the least distance from the tractor.

### 2.3. Selection Range of Control Points

#### 2.3.1. Key Point Coordinates

In order to improve the efficiency of path planning, obstacles with irregular contours are fitted with semicircles. As shown in Figure 5a, the global path is used as the dividing line, and the field can be divided into two areas: planted and unplanted. Next, we need to fit the contour of the obstacle in the working area with a semicircle to obtain the intersection points ($P_s$ and $P_e$) of the semicircle and the global path, as well as the coordinates of the vertex $P_v$ of the semicircle that envelops the obstacle [22].

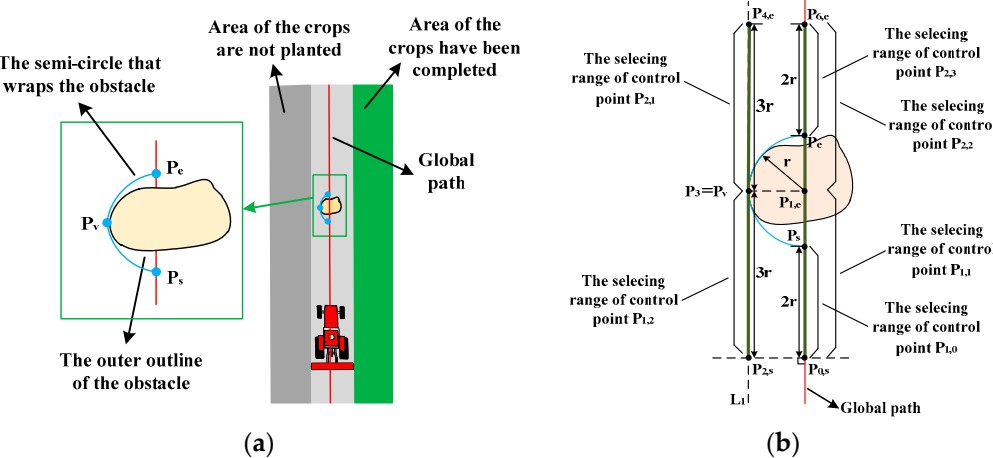

(**a**)                    (**b**)

**Figure 5.** Control point selection range. (**a**) Key point coordinates, (**b**) Control point selection range.

The obstacle avoidance path is calculated based on Equation (1), and it must be achieved in the rectangular coordinate system. Therefore, the WGS84 coordinate system is converted to a rectangular coordinate system by using Equation (14) in this paper, and then we can obtain the key points $P_s = (x_s, y_s)$, $P_e = (x_e, y_e)$, and $P_v = (x_v, y_v)$ in the rectangular coordinate system.

$$\begin{bmatrix} x_p \\ y_p \end{bmatrix} = \begin{bmatrix} A\cos(lat) * \cos(lon) \\ A\cos(lat) * \sin(lon) \end{bmatrix} \tag{14}$$

where (lon,lat) is the coordinate of point P in the WGS84 coordinate series, (xp,yp) is the coordinate of point P in the rectangular coordinate series, and A is defined as the following equation:

$$A = a^2 / \left( a^2\cos(lon)^2 + b^2\sin(lon)^2 \right)^{\frac{1}{2}} \tag{15}$$

where a is the semi-major axis of the earth ellipsoid and b is the semi-minor axis of the earth ellipsoid.

2.3.2. Control Point Selection Range

The geometric shape of the obstacle avoidance path is determined by control points, and the selection range of control points is shown in Figure 5b. The maximum distance between the tractor and global path is introduced by changing the coordinate of control point $P_{1,3}$, and we set the control point $P_{1,3} = P_y = (xy,yy)$ to obtain the minimum value.

From the properties of the third-order Bezier curve, it is known that the relationship between the positions of control points satisfies the following equation:

$$\frac{v_0}{|v_0|} // \frac{v_3}{|v_3|} // \frac{v_6}{|v_6|} // \overrightarrow{P_{1,0}P_{1,1}} // \overrightarrow{P_{1,2}P_{1,3}} // \overrightarrow{P_{1,3}P_{2,1}} // \overrightarrow{P_{2,2}P_{2,3}} \tag{16}$$

where $v_0$ is the speed vector when the tractor is in the control point $P_{1,0}$, $v_3$ is the speed vector when the tractor is in the control point $P_{1,3}$, and $v_6$ is the speed vector when the tractor is in the control point $P_{2,3}$.

Based on Equation (16), the selection range of control points $P_{1,0}$, $P_{1,1}$, $P_{2,2}$, and $P_{2,3}$ are all on the global path, and the selection range of control points $P_{1,2}$ and $P_{2,1}$ are all on the straight line $L_1$ passing through point $P_{1,3}$. In addition, the global path is parallel to the straight line $L_1$.

We can determine the coordinate of control point $P_{1,0}$ according to the straight line $P_{0,s}P_s$ with a length of 2r, and the selection range of control point $P_{1,0}$ shown in Equation (15) can be calculated based on Equation (16):

$$\begin{cases} \min\{2x_s - x_e, x_s\} \leq x_{1,0} \leq \max\{2x_s - x_e, x_s\} \\ y_{1,0} = \frac{y_e - y_s}{x_e - x_s}x_{1,0} + \frac{x_e y_s - x_s y_e}{x_e - x_s} \end{cases} \tag{17}$$

Similarly, it can be obtained that the selection range of control points $P_{2,3}$ is the straight line $P_eP_{6,e}$ with a length of 2r:

$$\begin{cases} \min\{2x_e - x_s, x_e\} \leq x_{2,3} \leq \max\{2x_e - x_s, x_e\} \\ y_{2,3} = \frac{y_e - y_s}{x_e - x_s}x_{2,3} + \frac{x_e y_s - x_s y_e}{x_e - x_s} \end{cases} \tag{18}$$

where $(x_{2,3}, y_{2,3})$ represents the coordinate of control point $P_{2,3}$, $(xs,ys)$ is the coordinate of key point $P_s$, and $(xe,ye)$ is the coordinate of key point $P_e$.

The selection range of control point $P_{1,1}$ is the straight line $P_{0,s}P_{1,e}$ with a length of 3r:

$$\begin{cases} \min\{2x_s - x_e, \frac{x_s + x_e}{2}\} \leq x_{1,1} \leq \max\{2x_s - x_e, \frac{x_s + x_e}{2}\} \\ y_{1,1} = \frac{y_e - y_s}{x_e - x_s}x_{1,1} + \frac{x_e y_s - x_s y_e}{x_e - x_s} \end{cases} \tag{19}$$

where $(x_{1,1}, y_{1,1})$ is the coordinate of control point P1,1, $(xs,ys)$ represents the coordinate of key point Ps, and $(xe,ye)$ is the coordinate of key point Pe.

The selection range of control point $P_{1,2}$ is the straight line $P_{2,s}$ Pv with a length of 3r:

$$\begin{cases} \min\{x_v + x_s - x_e, x_v\} \leq x_{1,2} \leq \max\{x_v + x_s - x_e, x_v\} \\ y_{1,2} = \frac{y_e - y_s}{x_e - x_s}x_{1,2} + \frac{x_e y_v - x_s y_v - x_s y_e + x_v y_v}{x_e - x_s} \end{cases} \tag{20}$$

where $(x_{1,2}, y_{1,2})$ is the coordinate of control point $P_{1,2}$, $(xv, yv)$ is the coordinate of key point $P_v$, $(xs,ys)$ is the coordinate of key point $P_s$, and $(xe,ye)$ is the coordinate of key point $P_e$.

The selection range of control point $P_{2,1}$ is the straight line $PvP_{4,e}$ with a length of 3r:

$$\begin{cases} \min\{x_v - x_s + x_e, x_v\} \leq x_{2,1} \leq \max\{x_v - x_s + x_e, x_v\} \\ y_{2,1} = \frac{y_e - y_s}{x_e - x_s}x_{2,1} + \frac{x_e y_v - x_s y_v - x_s y_e + x_v y_v}{x_e - x_s} \end{cases} \tag{21}$$

where $(x_{2,1}, y_{2,1})$ is the coordinate of control point $P_{2,1}$, $(xv, yv)$ is the coordinate of control point $P_v$, $(xs,ys)$ is the coordinate of key point $P_s$, and $(xe,ye)$ is the coordinate of key point $P_e$.

The selection range of control point $P_{2,2}$ is the straight line $P_{1,e}P_{6,e}$ with a length of $3r$:

$$\begin{cases} \min\left\{2x_e - x_s, \frac{x_s+x_e}{2}\right\} \leq x_{2,2} \leq \max\left\{2x_e - x_s, \frac{x_s+x_e}{2}\right\} \\ y_{2,2} = \frac{y_e - y_s}{x_e - x_s}x_{2,2} + \frac{x_e y_s - x_s y_e}{x_e - x_s} \end{cases} \tag{22}$$

where $(x_{2,2}, y_{2,2})$ is the coordinate of control point $P_{2,2}$, $(x_s,y_s)$ is the coordinate of key point $P_s$, and $(x_e,y_e)$ is the coordinate of key point $P_e$.

The selection range of all control points is shown in Equation (23):

$$\begin{cases} \min\{2x_s - x_e, x_s\} \leq x_{1,0} \leq \max\{2x_s - x_e, x_s\} \\ \min\left\{2x_s - x_e, \frac{x_s+x_e}{2}\right\} \leq x_{1,1} \leq \max\left\{2x_s - x_e, \frac{x_s+x_e}{2}\right\} \\ \min\{x_v + x_s - x_e, x_v\} \leq x_{1,2} \leq \max\{x_v + x_s - x_e, x_v\} \\ \min\{x_v - x_s + x_e, x_v\} \leq x_{2,1} \leq \max\{x_v - x_s + x_e, x_v\} \\ \min\left\{2x_e - x_s, \frac{x_s+x_e}{2}\right\} \leq x_{2,2} \leq \max\left\{2x_e - x_s, \frac{x_s+x_e}{2}\right\} \\ \min\{2x_e - x_s, x_e\} \leq x_{2,3} \leq \max\{2x_e - x_s, x_e\} \\ \begin{bmatrix} y_{1,0} & y_{62,3} \\ y_{1,1} & y_{52,2} \\ y_{1,2} & y_{42,1} \end{bmatrix} = \frac{1}{x_e - x_s}\left[(y_e - y_s)T_1 + T_2 T_3\right] \\ \begin{bmatrix} x_{1,3} & y_{1,3} \end{bmatrix} = \begin{bmatrix} x_v & y_v \end{bmatrix} \end{cases} \tag{23}$$

where $T_1$, $T_2$, and $T_3$ are defined as follows:

$$T_1 = \begin{bmatrix} x_{1,0} & x_{62,3} \\ x_{1,1} & x_{52,2} \\ x_{1,2} & x_{42,1} \end{bmatrix}, T_2 = \begin{bmatrix} y_s & -y_e & 0 & 0 \\ y_s & -y_e & 0 & 0 \\ y_v & -y_e & y_v & -y_v \end{bmatrix}, T_3 = \begin{bmatrix} x_e & x_e \\ x_s & x_s \\ x_v & x_v \\ x_s & x_s \end{bmatrix}$$

*2.4. Constraint Functions*

2.4.1. Geometric Constraints

The obstacle avoidance path has the following two geometric constraints. On the one hand, it is required that the curvature radius of the obstacle avoidance path should not be less than the minimum turning radius:

$$k(t) = \frac{\dot{x}\ddot{y} - \ddot{x}\dot{y}}{\left(\dot{x}^2 + \dot{y}^2\right)^{\frac{3}{2}}} \leq \frac{1}{R_{min}} \tag{24}$$

where $R_{min}$ is the minimum turning radius of the tractor.

On the other hand, the obstacle avoidance path should satisfy the anti-collision constraint described in Equation (25):

$$\sqrt{\left(x(t) - \frac{x_s + x_e}{2}\right)^2 + \left(y(t) - \frac{y_s + y_e}{2}\right)^2} \geq \frac{1}{2}\sqrt{(x_s - x_e)^2 + (y_s - y_e)^2} \tag{25}$$

where $(x_t, y_t)$ is any point of the obstacle avoidance path.

### 2.4.2. Tractor Kinematic Constraint

Due to the limitation of the mechanical structure, the turning angle of the tractor has a certain range. Therefore, it is required that the turning angle should not exceed the maximum turning angle.

$$\theta = \arctan\left(\frac{2L\sin\alpha}{L_d}\right) < \theta_{max} \tag{26}$$

where $L$ is the tractor wheelbase, $L_d$ is the distance between the tractor rear axle and preview point, $\alpha$ is the heading angle when the tractor arrives at the preview point, and $\theta_{max}$ is the maximum turning angle of tractor.

In addition, all constraints are shown in Equation (27):

$$\begin{cases} k = \dfrac{\dot{x}\ddot{y}-\ddot{x}\dot{y}}{\left(\dot{x}^2+\dot{y}^2\right)^{\frac{3}{2}}} \le \dfrac{1}{R_{min}} \\ \theta = \arctan\left(\dfrac{2L\sin\alpha}{l_d}\right) < \theta_{max} \\ \sqrt{\left(x(t) - \dfrac{x_s+x_e}{2}\right)^2 + \left(y(t) - \dfrac{y_s+y_e}{2}\right)^2} \ge \dfrac{1}{2}\sqrt{(x_s - x_e)^2 + (y_s - y_e)^2} \end{cases} \tag{27}$$

### 2.5. Objective Function

The objective function needs to minimize the error of tractor automatic navigation and maximize the land utilization. Therefore, m is designed as follows:

$$\begin{cases} m = w_1 d_{max} + w_2 \dfrac{4s'}{\pi\left[(x_s-x_e)^2+(y_s-y_e)^2\right]} \\ w1 + w2 = 1, (0 < w1 < 1, 0 < w2 < 1) \end{cases} \tag{28}$$

where m is the objective function value, $(x_s,y_s)$ is the coordinate of key point Ps, $(x_e,y_e)$ is the coordinate of key point Pe, and $d_{max}$ and s′ are defined as Equations (29) and (30), respectively.

$$d_{max} = \max\{d'_0, d'_1, \dots, d'_i, \dots\} \tag{29}$$

where d′i is the error value of the ith tractor path tracking simulation and d′i is the maximum error value in this tractor path tracking simulation.

$$\begin{aligned} s' &= S_{w1} + S_{w2} \\ &= w\left(\sqrt{(x_s - x_0)^2 + (y_s - y_0)^2} + \sqrt{(x_e - x_6)^2 + (y_e - y_6)^2}\right) \end{aligned} \tag{30}$$

where $S_{W1}$ is the area of area $W_1$ as in Figure 6, $S_{W2}$ is the area of area $W_2$ as shown in Figure 6, s′ is the total area of wasteful land, $(x_s,y_s)$ represents the coordinate of key point Ps, $(x_e,y_e)$ is the coordinate of key point Pe, $(x_{1,0},y_{1,0})$ is the coordinate of control point $P_{1,0}$, and $(x_{2,3},y_{2,3})$ is the coordinate of control point $P_{2,3}$.

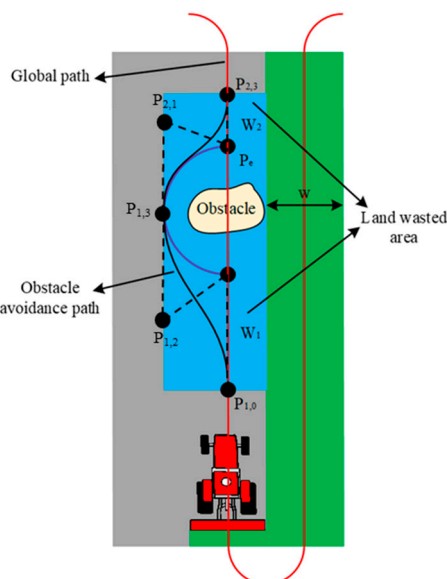

**Figure 6.** An example of wasteful land.

### 2.6. Search Strategy of Optimal Control Points

In this paper, GA is used to search for the optimum control points, and the search strategy of optimal control points consists of two processes [23]. First, the individual and population are established using Equation (23), and the fitness function is established using Equation (26). Secondly, the kinematic model of the tractor is combined with GA to search for obstacle avoidance paths that meet the constraints of Equation (26).

#### 2.6.1. Individual Establisher

In GA, an individual is expressed by $Si = \{P_{1,0}, P_{1,1}, P_{1,2}, P_{1,3}, P_{2,0}, P_{2,1}, P_{2,2}, P_{2,3}\} \in \Omega$, where $\Omega$ is the search space [24,25]. Each individual consists of discrete units called genes, and genes are the control point coordinate in this paper. The control points and Equation (1) are used to generate the obstacle avoidance path. Then, the obstacle avoidance path of the WGS84 coordinate system is obtained using Equation (31):

$$\begin{bmatrix} \text{lon} \\ \text{lat} \end{bmatrix} = \begin{bmatrix} \arcsin\left(\frac{y}{\sqrt{x^2+y^2}}\right) \times \frac{180}{\pi} \\ \arccos\left(\frac{\sqrt{x^2+y^2}}{A}\right) \times \frac{180}{\pi} \end{bmatrix} \tag{31}$$

where (lon,lat) is the coordinate of a point in the WGS84 coordinate system, (x,y) is the coordinate of a point in the rectangular coordinate system, and A is defined as follows:

$$A = a^2 / \left(a^2\cos(\text{lon})^2 + b^2\sin(\text{lon})^2\right)^{\frac{1}{2}} \tag{32}$$

where a is the semi-major axis of the earth ellipsoid and b is the semi-minor axis of the earth ellipsoid.

#### 2.6.2. Fitness Function

The fitness function is established according to the objective function shown in Equation (25) and constraint functions shown in Equation (33).

$$f = \frac{X}{m} = \frac{X}{\left(d_{\max} + \frac{4s'}{\pi\left[(x_s-x_e)^2+(y_s-y_e)^2\right]}\right)} \tag{33}$$

where f is the fitness function value and X is defined as follows:

$$x = \begin{cases} 1 \text{ if obstacle avoidance path satisfy the Equation (24)} \\ 0 \text{ else} \end{cases} \tag{34}$$

## 3. Results and Discussion

### 3.1. Experimental Settings

To evaluate the algorithm proposed in this paper, we designed an experiment with the parameters in Table 1. First, we set up a semicircle with a radius of 3 m to envelop the obstacle in the field. Then, the key point coordinates and radius of the semicircle were input into GA to calculate the optimum obstacle avoidance path. In addition, the population size was set to 80, the mutation probability was 0.7, and the crossover probability was 0.64. The number of iterations affected the result and computation time. Therefore, iterations will be terminated when the average fitness of the population exceeds 10 times at a constant value. Finally, the tractor automatic navigation error and wasted land area were obtained. The results obtained by the algorithm in references and this paper are compared.

**Table 1.** Parameters of obstacle avoidance path planning.

| Parameter | Value |
|:---:|:---:|
| $d_1$ | 2.5 m |
| $w$ | 2.4 m |
| $L$ | 2.06 m |
| $R_{min}$ | 4 m |
| $\theta_{max}$ | $\pi/6$ |

Note: $d_1$ is the distance from the point on the planned path that is the smallest distance from the tractor to the preview point on the path, $w$ is the width of the job line, $L$ is the tractor wheelbase, $R_{min}$ is the minimum turning radius of the tractor, $\theta_{max}$ is the maximum turning angle of tractor.

This paper used the modified Dongfanghong LF1104-C tractor with the automatic navigation function as the experimental platform, as shown in Figure 7. Dongfanghong has remote control and positioning and navigation functions. It is equipped with an automatic steering system, tractor control system, radar and visual measurement system, remote video transmission system, monitoring system, remote control system, and other information and control systems [26]. The specific parameters of the tractor are shown in Table 2.

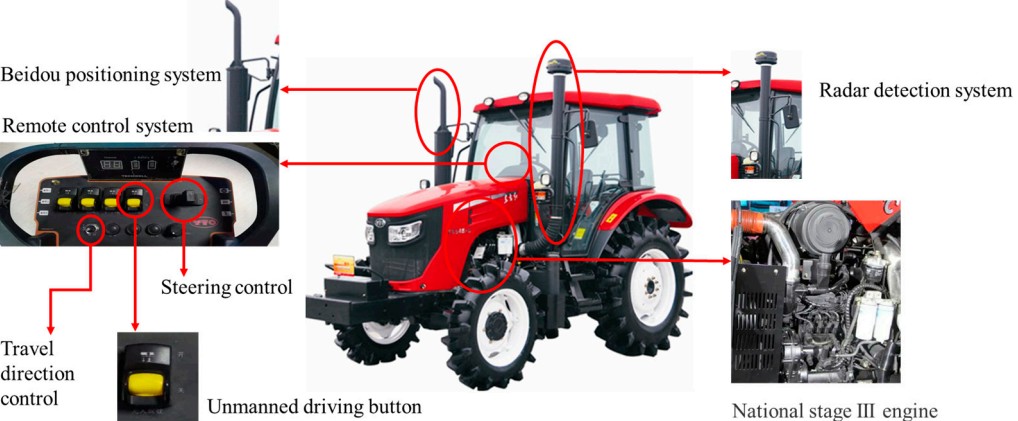

**Figure 7.** Dongfanghong LF1104-C.

**Table 2.** Dongfanghong LF1104-C basic parameters.

| Model | LF1104-C |
|---|---|
| Type | Four-Wheel Drive |
| PTO Power (Kw) | $\geq$67 |
| Maximum Traction (Kn) | 19.8 |
| Length (Including Suspension) | 4436 |
| Width (Factory Wheelbase) | 2250 |
| Height | 2765 |
| Wheelbase (Mm) | 2314 |
| Front Wheel (Factory Wheelbase) | 1748–2000 (1760) |
| Rear Wheel (Factory Wheelbase) | 1620–2120 (1632) |
| Ground Clearance (Mm) | 440 (Under The Bent Rod Housing) |
| Front Counterweight (Kg) | 400 |
| Rear Counterweight (Kg) | 300 |
| Minimum Operation Mass (Assembly Frame Without Counterweight, With Cab (Kg)) | 4250 |

At the same time, an industrial computer with a Windows 10 operating system and a corn planter with a width of 2.4 m were installed on the tractor. The software of the whole experimental system was developed based on Python, the hardware was controlled by a single chip microcomputer, and the CAN bus was used for data transmission between software and hardware [27]. The experiment was conducted at the Anhui Agricultural University Wanbei comprehensive experimental station, where the field was flat enough for us to set-up artificial obstacles.

The obstacle in Figure 8a is not within the operating range of the agricultural machine, which remains in normal operating condition. In Figure 8b, the LIDAR device on the agricultural machine detects the presence of obstacles in the current operation line and starts the obstacle avoidance operation of the current operation line, the obstacle avoidance operation is shown in Figure 8c to Figure 8i. When the farm machine bypasses the obstacle, the end point of the third-order Bessel curve will cause the farm machine to return to the original operation row, as in Figure 8i, and the farm machine will continue to drive along the original planning path, as in Figure 8j.

*3.2. Obstacle Avoidance Path Planning*

The coordinates of key points collected through BeiDou positioning equipment are shown in Table 3. The results of the average population fitness value are shown in Figure 9. It can be noticed from the figure that the highest average population fitness value of the experiment was 4.09597 at the 20th iteration. In the experiment, the iterations were terminated when we reached the point that increasing the number of iterations by more than 10 no longer affected the result. In our result, the point was 20. Therefore, we obtained the optimal solution when the number of iterations was 30.

The algorithm of obstacle avoidance path planning with the objective of curvature continuity proposed by Xi et al. is a popular method in recent years [5]. Lee et al. proposed a novel obstacle avoidance algorithm for mobile robots based on finite memory filtering (FMF) [28]. So, we chose these to compare the performance of the algorithm proposed in this paper. The results of the obstacle avoidance path are shown in Figure 10. There are four obstacle avoidance paths in Figure 10; one path is the obstacle avoidance path planned by the algorithm in this paper (TA). The other three comparison algorithms are the obstacle avoidance path planned by the algorithm of Xi et al. (PR), the obstacle avoidance path planned by the algorithm of Lee (DWA), and the artificial potential field method (APF).

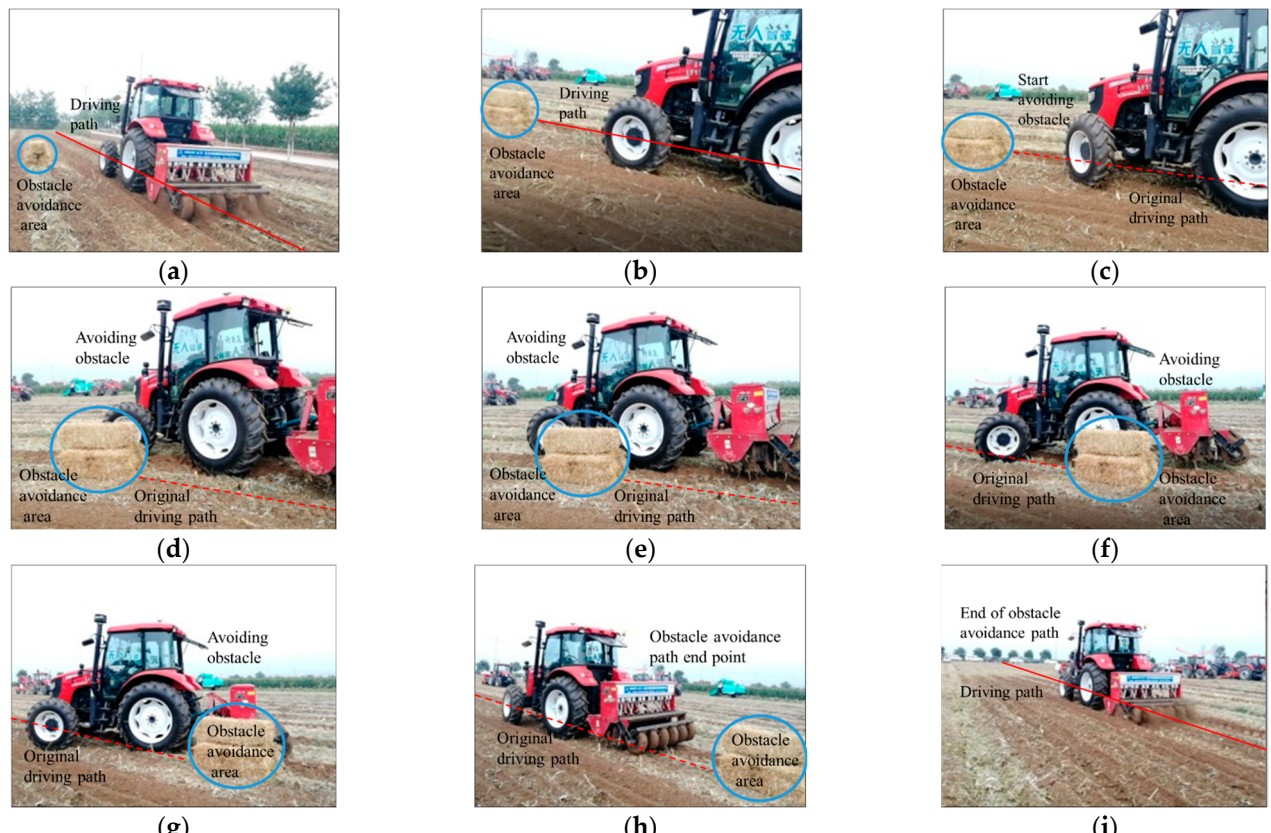

**Figure 8.** Experimental platform and experimental site.

**Table 3.** Longitude and latitude coordinates of key points.

| Key Points | Coordinates |
|:---:|:---:|
| $p_s$ | (117.07882885798031, 33.69548283272718) |
| $p_v$ | (117.0787965, 33.69550988) |
| $p_e$ | (117.07882885798195, 33.69553692727986) |

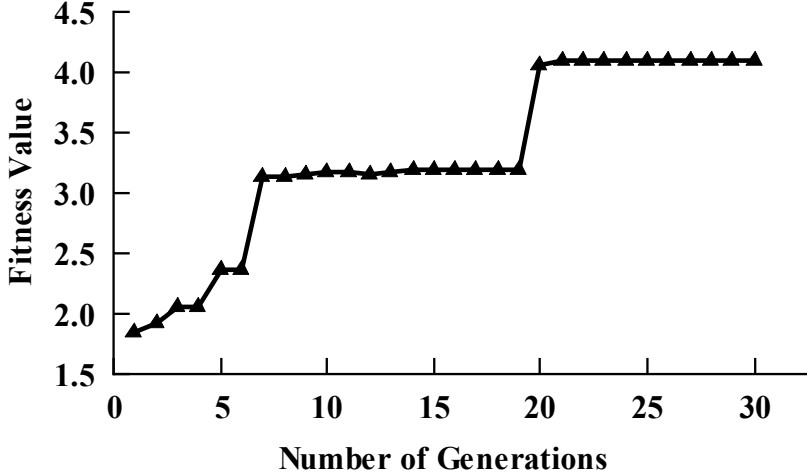

**Figure 9.** Result diagram of average population fitness value in GA.

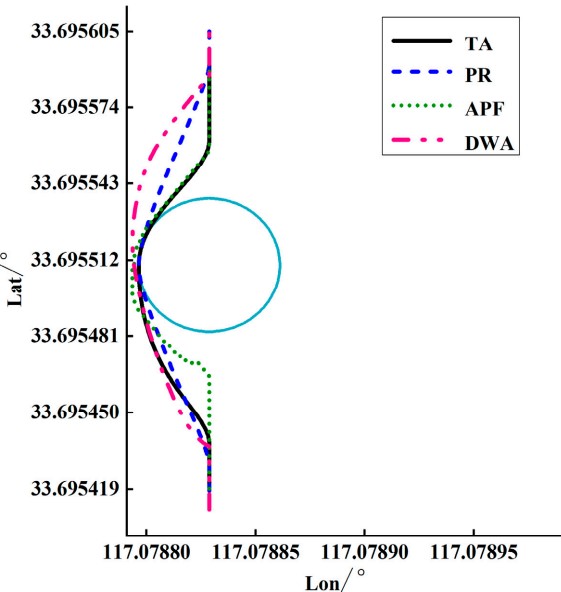

**Figure 10.** Obstacle avoidance path planning by different algorithm.

　　The smoothness of the obstacle avoidance path affects the path tracking accuracy. The smoother the path, the smaller the curvature, and the smaller the curvature, the higher the tractor path tracking accuracy. The results of the comparison between TA and PR in terms of maximum curvature, average curvature, and wasted land area are presented in Table 4. As shown in Table 4, the curvature value of TA was much smaller than those of the other three algorithms, and the land wasted area of TA was also much smaller than those of PR and DWA. Although the land wasted area was smaller for APF, the curvature value for APF was too large and more demanding for the steering operation of the farm machinery. Therefore, TA was more advantageous than the other three algorithms.

**Table 4.** The data of path curvature and wasted land area with different path planning methods.

| Path | $k_{max}$ | $k_{ave}$ | $S_p/m^2$ |
|:---:|:---:|:---:|:---:|
| TA | 0.66 | 0.17 | 23.14 |
| PR | 1.71 | 0.12 | 36.00 |
| APF | 4.73 | 0.79 | 22.61 |
| DWA | 1.82 | 0.13 | 38.17 |

Note: $k_{max}$ is the maximum curvature, $k_{ave}$ is the mean curvature, $S_p$ is wasteful land area.

### 3.3. Field Experiment

　　We conducted a field experiment to verify the performance of the algorithm in this paper. In the experiment, the initial position of the tractor was set to deviate from the obstacle avoidance path by 0.01 m, and the initial wheel angle and speed of the tractor were 0° and 2 m/s, respectively. The results are shown in Figures 11 and 12.

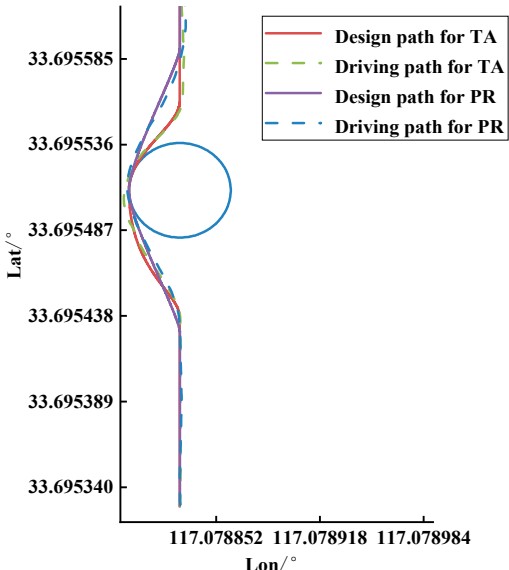

**Figure 11.** The result of path tracking.

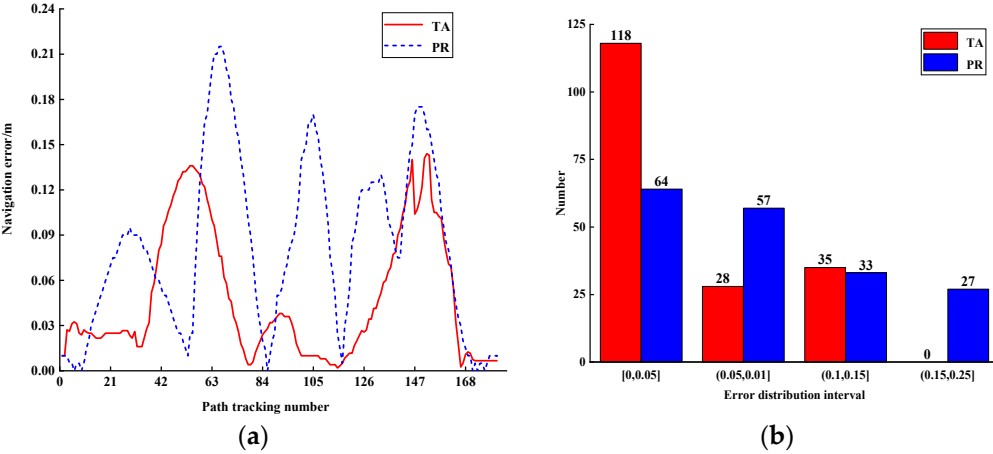

**Figure 12.** Navigation error of tractor path tracking. (**a**) Navigation error of tractor path tracking, (**b**) A navigation error comparison between TA and PR.

The path tracking results for TA and PR are shown in Figure 11. The navigation errors are illustrated in Figure 12. From Figure 12a, it can be observed that the maximum navigation error and the average navigation error of TA were 0.144 m and 0.048 m, respectively. The maximum navigation error and the average navigation error of PR were 0.215 m and 0.082 m, respectively. The maximum navigation error and the average navigation error of TA were 33% and 41.5% lower than those of PR, respectively. The navigation errors obtained from the statistical experiments and the final results are shown in Figure 12b.

From Figure 12b, it can be observed that more than 50% of TA navigation errors were less than 0.05 m, while only 35.4% of PR errors were less than 0.05 m. TA errors were less than 0.15 m, while 14.9% of PR errors were larger than 0.15 m. In short, compared with the current mainstream obstacle avoidance path planning algorithms, the algorithm proposed in this paper plans a smoother obstacle avoidance path and can further improve the tractor path tracking accuracy.

## 4. Conclusions

In this paper, a new method for the tractor obstacle avoidance path planning is proposed. The third-order Bezier curve is used to plot the obstacle avoidance path. Therefore, in this paper, the optimal obstacle avoidance path is generated by searching for the optimal

control points. The key point coordinates of the obstacle avoidance path are determined by the global path and obstacle information, and the control point selection range is determined by the key point coordinates. GA is used to search for the optimal control points from the control point selection range and the optimal obstacle avoidance path satisfying the tractor obstacle avoidance constraint, the minimum turning radius constraint, and the tractor kinematic constraint generated by the optimal control point. The algorithm proposed in this paper is superior to some existing methods in terms of navigation accuracy and wasted land area for the planned path.

**Author Contributions:** Writing—original draft preparation, H.C.; writing—review and editing, H.X.; validation, L.S.; supervision, T.S. All authors have read and agreed to the published version of the manuscript.

**Funding:** This research received no external funding.

**Institutional Review Board Statement:** Not applicable.

**Data Availability Statement:** Not applicable.

**Conflicts of Interest:** The authors declare no conflict of interest.

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
