# Peer review of "Research on Tractor Optimal Obstacle Avoidance Path Planning for Improving Navigation Accuracy and Avoiding Land Waste"

_agriculture, doi:10.3390/agriculture13050934_

Round 1
Reviewer 1 Report
In this paper, an obstacle avoidance path planning method based on genetic algorithm (GA) and Bezier curve is proposed. Authors proposed Bezier curve to plot the obstacle avoidance path, and the control points that determine the geometry of the third-order Bezier curve based on the global path and obstacle locations. This is an important issue with regard to maximise the use of the area of land allocated for cultivation.
I am afraid that the article presented for review needs to be improved in many aspects. Firstly, the authors have 30 references but failed to refer to 12 of them in the text! The text needs to be re-checked for the required justification. For example, there is an unnecessary paragraph on line 50, and many figures are aligned once to the right, once to the left edge or to the centre. Also missing spaces between words as in line 278.
I believe that the most important methodological error is to select only one algorithm for comparison. The authors claim to have chosen the most popular algorithm for comparison, which does not necessarily mean that other algorithms are not superior to the presented one in terms of reducing the loss of cultivated field area. This needs to be improved.
Author Response
Dear reviewer,please see the attachment.

Reviewer 2 Report
1. It is one thing to detour around an obstacle with a unit. Another thing is how it then continues to work, performing adjacent passes. This is especially important when sowing row crops. Therefore, if the authors do not show in detail and clearly how to carry out the work of the robot-unit after bypassing obstacles in agricultural fields, then this article should be rejected from publication in this journal. It probably belongs in some mathematical journal.
2. How can one obtain the second and third equations of system (4) by looking at Fig. 3?
3. Why did the article's authors stop using a third-order Bezier curve?
Author Response

(The authors gave the same response as above.)

Reviewer 3 Report
The paper discusses optimal obstacle avoidance of agricultural tractor to improve land waste and navigation accuracy.
The paper is well-written a interesting for broader audience of agricultural engineers.
My comments and questions are as follows.
Line 226 Objective function
It is difficult to comprehend the definition of land waste. Please explain the definition of land waste in detail here.
I think there is trade-off between navigation accuracy and land waste. So, you should introduce weight coefficient to objective function formula (21) like following.
m=w1*dmax+w2*4s'/(pi*(xs-xe)^2+(ys-ye)^2)
w1+w2=1, (0<w1<1, 0<w2<1)
If you think navigation accuracy is more important than land waste, w1 become larger.
I think optimal path could be different under different weight coefficients.
Line 268 Experimental settings
You conducted field experiments directly.
These experiment results are interesting.
If you conduct some computer simulation using vehicle dynamics simulator (CarSim, CarMaker, or dynamic model using MATLAB), the interoperation of field experiments become more interesting.
Please consider to conduct computer simulation model.
Line 114 2.2 Tractor kinematic model
You assume static model in this chapter.
But dynamic bicycle model is popular to use like this.
(Towards agrobots: Identification of the yaw dynamics and trajectory tracking of an autonomous tractor - ScienceDirect)
Please explain the reason why you choose static model in this research.
Author Response

(The authors gave the same response as above.)

Round 2
Reviewer 2 Report
My comments are in green.
Sincerely,
Volodymyr Nadykto

Author Response
Dear Reviewer,Please see the attachment!
